# The Phylogeny and Taxonomy of *Cryptothecia* (Arthoniaceae, Ascomycota) and *Myriostigma* (Arthoniaceae, Ascomycota), including Three New Species and Two New Records from China

**DOI:** 10.3390/jof10040274

**Published:** 2024-04-09

**Authors:** Junxia Xue, Yutong Cai, Lulu Zhang

**Affiliations:** Institute of Environment and Ecology, Shandong Normal University, Jinan 250300, China; 2021021173@stu.sdnu.edu.cn (J.X.); 2022021206@stu.sdnu.edu.cn (Y.C.)

**Keywords:** lichenized fungi, crustose thallus, diversity, identification key, phylogenetic analysis

## Abstract

*Cryptothecia* and *Myriostigma* are important elements of crustose lichen communities in tropical to subtropical forests, but little research has been done on these two genera in China. Morphological and molecular phylogenetic approaches to investigate species diversity of *Cryptothecia* and *Myriostigma* from Southern China were carried out in this study. We find five species of *Cryptothecia* and *Myriostigma* in our study, including three new species (*M. flavescens*, *M. hainana* and *M. laxipunctata*) and two new records (*C. bartlettii* and *C. inexspectata*). In addition, a phylogenetic tree based on mtSSU, RPB2 and nLSU illustrates the placement of the five species and supports the delimitation of the three new taxa. Detailed descriptions of morphological, ecological and chemical characteristics and illustrations are provided for every species. A key to all known Chinese *Cryptothecia* and *Myriostigma* species is also provided.

## 1. Introduction

*Cryptothecia* Stir. is a genus of lichenized fungi belonging to Arthoniaceae, Arthoniales, Arthoniomycetes, Ascomycota [1]. The genus *Cryptothecia* is recognized by corticolous, ecorticate, greenish grey to whitish grey thalli; whitish byssoid prothalli; a medulla with calcium oxalate crystals; trentepohlioid photobiont, yellowish green; broadly clavate to globose, thick-walled asci loosely dispersed on the thallus or closely aggregated within ascigerous areas; and muriform ascospores [2]. Almost all *Cryptothecia* species are fertile, except *C. isidioxantha* Aptroot & M. Cáceres, *C. duplofluorescens* Aptroot & Souza, *C. reagens* Aptroot & Schumm, and *C. lecanorosorediata* Aptroot & M. Cáceres which are not known to produce ascospores [3,4,5,6].

Recently, phylogenetic analyses revealed that *Cryptothecia* was polyphyletic and *Myriostigma* Krempelhuber (1874) was resurrected because the type species *Myriostigma candidum* Kremp. [=*Cryptothcia candida* (Kremp.) R. Sant.] represents a lineage separate from *Cryptothecia*. *Myriostigma* species are characterized by raised maculate ascomata-like ascigerous areas; eight-spored, thick-walled, globular and stalked asci; and bean-shaped, muriform ascospores [7].

*Cryptothecia* and *Myriostigma* are distributed in tropical and subtropical humid forests. To date, approximately 113 species are known worldwide [6,8,9,10,11,12,13,14,15,16], and only 6 species have been reported in China [17,18]. During our research on these two genera, three new species (*M. flavescens* J.X. Xue & Lu L. Zhang, *M. hainana* J.X. Xue & Lu L. Zhang and *M. laxipunctata* J.X. Xue & Lu L. Zhang) and two new records (*C. bartlettii* G. Thor and *C. inexspectata* G. Thor) were found.

## 2. Materials and Methods

### 2.1. Morphology and Anatomy

The specimens examined were collected in Fujian, Guangxi, Hainan and Yunnan Provinces, China, and preserved in the Lichen Section of Botanical Herbarium (SDNU, Shandong Normal University). External morphological features were studied with a dissecting microscope (COIC XTL7045B2) (Olympus, Toyko, Japan), and photos were taken under an Olympus SZX16 (Olympus, Toyko, Japan) with DP72. Internal anatomical features were observed and measured by a microscope (Olympus CX41) (Olympus, Toyko, Japan), and images were taken with an Olympus BX61 (Olympus, Toyko, Japan) with DP 72.

### 2.2. Colour Reaction

The thallus and medulla colouring reactions were carried out by P (a saturated solution of p-phenylenediamine in 95% ethyl alcohol) (Tianjin Fuyu Fine Chemical Co., Tianjin, China), C (a saturated solution of aqueous sodium hypochlorite) (Tianjin Fuyu Fine Chemical Co., Tianjin, China), K (a 10% aqueous solution of potassium hydroxide) (Tianjin Damao Chemical FTY., Tianjin, China), I (a 3% solution of Lugol’s iodine) (Tianjin Fuyu Fine Chemical Co., Tianjin, China), and long-wavelength UV light. Polarized light microscopy (pol) was used to locate crystals in the thallus sections. Then, H_2_SO_4_ (a 25% solution of sulfuric acid) (Yantai Yuandong Fine Chemical Co., Yantai, China), was used to test whether the characteristic needle-shaped crystals formed when calcium oxalate was present in the thallus.

### 2.3. Chemical Analysis

The lichen substances were identified using standardized thin-layer chromatography (TLC). The particularly stable and reliable solvent C (toluene/acetic acid 170:30) was used in this study as it often provides the best discrimination of lichen substances. In this study, *Lethariella cladonioides* (Nyl.) Krog. containing atranorin and norstictic acid was used as the partition standard sample.

We used a pencil, carefully drawing a straight line 1.5 cm from the bottom of the glass silicone board. A point was marked every 1 cm on the straight line, which was the sample point. We soaked the lichen fragments in c. 1 mL of acetone for 30 min in a small test tube. Then, we used microcapillary tubes to sample them separately according to the position of the sampling points on the glass silicone board. After sampling, we placed the silicone board in a chromatography cylinder and placed it 1 cm below the solvent level. When the leading edge of the solvent moved from the origin to about 1 cm from the top of the silicone board, we removed the silicone board and dried the solvent on the board surface with a hair dryer. The silicone board was dried and then examined under short wavelength (254 nm) ultraviolet light for pigments. Later, it was sprayed with 10% sulfuric acid and heated at 100 °C in an oven for 10 min to develop the spots. The Rf values and color of each lichen substance were recorded and immediately examined under long wavelength (365 nm) ultraviolet light. The Rf values, as well as the fluorescent properties, were compared and analyzed to confirm the identity of the substance [19,20].

### 2.4. DNA Extraction, PCR Amplification and Sequencing

DNA was extracted directly from the clean growing portions of the thalli of recently collected specimens. Genomic DNA was extracted using the Sigma-Aldrich REDExtract-N-Amp Plant PCR Kit (St. Louis, MO, USA) following the manufacturer’s instructions, except for the use of only 30 μL of extraction buffer and 30 μL of dilution buffer.

In our study, we found that the ITS sequence, typically considered crucial for distinguishing between lichen species, is essentially identical in *Cryptothecia* and *Myriostigma* and cannot be used as a standard for distinguishing between species within these two genera. Therefore, three other loci were amplified: the mtSSU gene with primer pairs mtSSU1 and mtSSU3R [21], the RPB2 gene with RPB2-7cF and RPB2-11aR [22], and the nLSU gene with LIC24R and LR7 [23,24]. The 50 μL PCR mixture consisted of 2 μL of DNA, 2 μL of each primer, 25 μL of 2 × Taq PCR MasterMix [Taq DNA Polymerase (0.1 unit/μL); 3 mM MgCl_2_; 100 mM KCl; 0.5 mM dNTPs; and 20 mM Tris-HCl (pH 8.3)] (Tiangen, Beijing, China), and 19 μL of dd H_2_O. The PCR conditions for mtSSU amplification were set for initial denaturation at 94 °C for 10 min; 34 cycles of 95 °C for 45 s, 50 °C for 45 s, and 72 °C for 90 s; and a final extension at 72 °C for 10 min. The PCR conditions for RPB2 amplification were set for an initial denaturation at 94 °C for 10 min; 34 cycles of 94 °C for 45 s, 52 °C for 50 s, and 72 °C for 1 min; and a final extension at 72 °C for 5 min. The PCR conditions for nLSU amplification were set for an initial denaturation at 95 °C for 15 min; 45 cycles of 95 °C for 45 s, 53 °C for 45 s, and 72 °C for 1 min; and a final extension at 72 °C for 7 min. Sequencing was performed by BioSune Biological Technology (Shanghai, China).

### 2.5. Sequence Alignment and Phylogenetic Analysis

To ascertain that all the new sequences were reliable, the newly generated sequences were compared to the sequences available in the GenBank database (http://www.ncbi.nlm.nih.gov/BLAST/, accessed on 10 September 2023), and all the new sequences were assembled using SeqMan v.7.0 (DNAstar packages). The sequences were aligned and edited using the online version of MAFFT v.7.0.26 and MEGA v.7.0. The algorithm of MAFFT chooses automatically (FFT-NS-1, FFT-NS-2, FFT-NS-i or L-INS-i; depending on the data size). The species *Chiodecton natalense* Nyl. was chosen as the outgroup [12].

Phylogenetic relationships were inferred using maximum likelihood (ML) and Bayesian inference (BI). The three gene matrices were combined by Geneious Prime 2023.0.4. We used the CIPRES Science Gateway (http://www.phylo.org/portal2/, accessed on 1 March 2024) [25] and performed ML analyses using RaxML-HPC v. 8.2.12 [26] under the default parameters as implemented on CIPRES, and support values were based on 1000 nonparametric bootstrap pseudoreplicates. We used PhyloSuite [27] to infer BI phylogenies via MrBayes 3.2.6 [28] under a partition model, for which the initial 25% of the sampled data were discarded as burn-in. The stationarity of the analysis was determined by examining the standard deviation of the split frequencies (<0.01). Bootstrap support (BS) ≥ 70 and posterior probability (PP) ≥ 0.95 were considered significant support values. The phylogenetic trees generated were visualized with FigTree v. 1.4.2 [29].

## 3. Results

### 3.1. Phylogenetic Analyses

A total of 10 mtSSU sequences, 7 RPB2 sequences and 2 nLSU sequences were newly generated from 10 specimens. We constructed ML and BI topologies based on these mtSSU, RPB2 and nLSU sequences and 87 additional sequences downloaded from NCBI (https://www.ncbi.nlm.nih.gov/, accessed on 25 February 2024) (Table 1). The phylogenetic trees obtained from ML and BI analyses exhibited similar topologies; therefore, we present only the ML tree, with BS ≥ 70 for the ML analysis and PP ≥ 0.95 for the Bayesian analysis (Figure 1).

The combined sequence matrices revealed three new monophyletic lineages corresponding to three new species: *Myriostigma flavescens* J.X. Xue & Lu L. Zhang, sp. nov.; *M. hainana* J.X. Xue & Lu L. Zhang, sp. nov.; and *M. laxipunctata* J.X. Xue & Lu L. Zhang, sp. nov. All the species positions had strong support in the ML and Bayesian analyses. In our phylogenetic tree, we can see those three new species (*M. flavescens*, *M. hainana* and *M. laxipunctata*) are clustered with *Myriostigma* and *Stirtonia.* According to Aptroot [30], *Stirtonia* had transversely septate ascospores; thus, based on morphological characters and phylogenetic analysis, we propose three new species in *Myriostigma.* This is in addition to two new Chinese records, as both are clustered with *Cryptothecia subnidulans* (the type species of *Cryptothecia*), which supported the delimitation of two new Chinese records.

### 3.2. Taxonomy

#### 3.2.1. The New Species

*Myriostigma flavescens* J.X. Xue & Lu L. Zhang, sp. nov. (Figure 2)

MycoBank No: 850048

Diagnosis: The new species differs from other species of *Myriostigma* by ascigerous areas with brown dots and erumpent angular to irregular outlines, yellow ascospores (58–76 × 19–28 µm), gyrophoric acid and confluentic acid as secondary metabolites.

Type: CHINA, Yunnan Province, Xishuangbanna Dai Nationality Autonomous Prefecture, Jinghong City, Primitive Forest Park, 22°1′55.75″ N, 100°52′37.47″ E, alt. 689 m, on bamboo, 7 March 2023, L.L. Liu et al. 20230612 (SDNU, holotype).

GenBank accession numbers: SSU PP051268, PP051267; RPB2 PP130144.

Description: Thallus corticolous or bambusicolous, up to 3 cm in diameter, ecorticate, cottony, dull, greenish grey, loosely attached to the substrate, 50–70 µm thick. Isidia not observed. Prothallus usually distinct, linear shaped, dark brown to black. Medulla white, with calcium oxalate crystals. Photobiont trentepohlioid, cells rounded to elliptical, single or a few cells aggregated, 6–17 × 6–13 µm. Hyphae 1.5–2 µm wide.

Ascigerous areas distinct, generally delimited, erumpent, slightly raised above the thallus level, angular to irregular in outline, 1–5 mm diam. and 115–165 µm tick, white with dense brown dots indicate individual asci. Asci pale yellow, frequent, fragile, globose to subglobose, often covered by hyaline hyphae, 8-spored, 95–100 × 83–95 µm. Ascospores yellow, oblong, muriform, curved, often broader in the centre, 58–76 × 19–28 µm.

Pycnidia not observed.

Chemistry: thallus C+ red, K–, P–, UV+ pale grey-white; medulla and paraphysoids I+ sky-blue. TLC: gyrophoric acid and confluentic acid.

Etymology: The epithet refers to its yellow ascospores.

Ecology and distribution: This species is found only in China on bamboo and trees in a humid tropical forest in Yunnan Province.

Notes: Morphologically, *Myriostigma flavescens* is similar to *M. candidum* Kremp. in having irregular erumpent whitish ascigerous areas, but *M. candidum* has smaller ascospores (40–65 × 12–25 µm) and 2′-*O*-methylanziaic acid and 2′-*O*-methylperlatolic acid as secondary metabolites [31]. In addition, *Myriostigma flavescens* is similar to *M. filicinum* (Ellis & Everh.) Frisch & G. Thor, but the latter has rounded to somewhat irregular outlines of ascigerous areas, smaller asci (60–90 × 50–70 µm) and perlatolic acid as a secondary metabolite [31].

Additional specimens examined: CHINA, Yunnan Province, Xishuangbanna Dai Nationality Autonomous Prefecture, Jinghong City, Primitive Forest Park, 22°1′55.75″ N, 100°52′37.47″ E, alt. 689 m, on bamboo, 7 March 2023, L.L. Liu et al. 20230641 (SDNU); ibid., 20230623 (SDNU); ibid., 22°2′9.71″ N, 100°53′5.81″ E, alt. 746 m, on the bark of trees, 7 March 2023, L.L. Liu et al. 20230695 (SDNU).

**Figure 2 jof-10-00274-f002:**
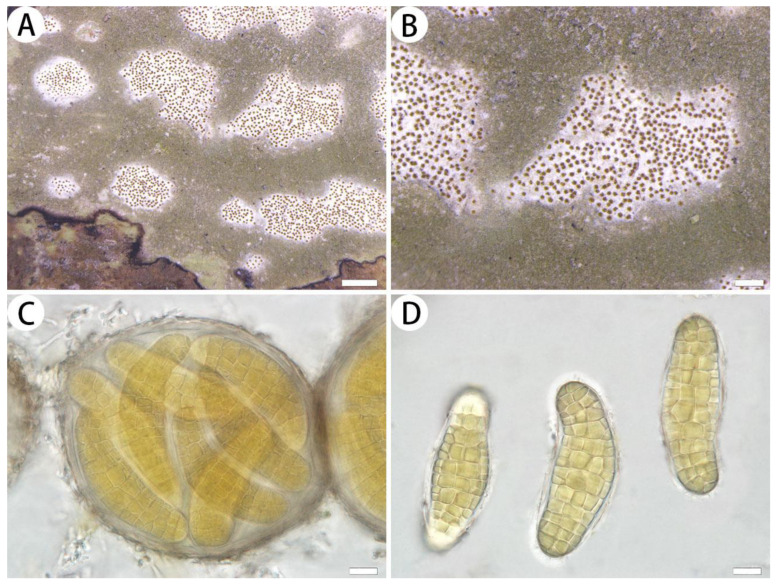
*Myriostigma flavescens* (SDNU 20230612, holotype). (**A**) Thallus and prothallus. (**B**) Ascigerous areas. (**C**) Asci. (**D**) Ascospores. Scale bars: 500 µm (**A**), 200 µm (**B**), 10 µm (**C**,**D**).

*Myriostigma hainana* J.X. Xue & Lu L. Zhang, sp. nov. (Figure 3)

MycoBank No: 849419

Diagnosis: The new species differs from other species of *Myriostigma* by having I– medulla, gyrophoric acid and methyl 2′-*O*-methylmicrophyllinate as secondary metabolites.

Type: CHINA, Hainan Province, Wenchang City, Beigang Village, 19°32′46.30″ N, 110°51′10.44″ E, alt. 26 m, on the bark of trees, 7 March 2023, J.X. Xue & Y.M. Zhang 20230061 (SDNU, holotype).

GenBank accession numbers: SSU PP051271, PP051272; RPB2 PP101845, PP109365.

Description: Thallus corticolous, up to 4 cm in diameter, ecorticate, cottony, dull, greenish grey to whitish grey, loosely attached to the substrate, 100–162 µm thick. Isidia not observed. Prothallus usually distinct, thin, whitish byssoid, mainly composed of interwoven hyphae, 0.5–1 mm wide, forming a dark brown line while bordering different species. Medulla white, with calcium oxalate crystals. Photobiont trentepohlioid, cells elliptical to oblong, single or a few cells aggregated, 9–15 × 6–9 µm. Hyphae 1–2 µm wide.

Ascigerous areas indistinct, not delimited. Asci frequent, fragile, globose to subglobose, scattered and immersed across the thallus, often covered by hyaline hyphae, 8-spored, 120–138 × 120–135 µm; sometimes 1–6(–7) spores aborted, thus, these asci remain (1–)2–7-spored. Ascospores hyaline, pale yellow when mature, oblong, muriform, straight, often broader in the centre, 52–88 × 24–47 µm.

Pycnidia not observed.

Chemistry: thallus C+ red, K–, P–, UV–; medulla and paraphysoids I–. TLC: gyrophoric acid and methyl 2′-*O*-methylmicrophyllinate.

Etymology: The epithet refers to the discovery of this new species in Hainan Province, China.

Ecology and distribution: This species is found only in China on the bark of trees in a humid tropical forest in Hainan Province.

Notes: Morphologically, *Myriostigma hainana* is the first species of the genus *Myriostigma* that contains 2′-*O*-methylmicrophyllinate. It is similar to *Cryptothecia subtecta* Stirt. in having indistinct ascigerous areas, but *C. subtecta* has an I+ blue medulla, smaller ascospores (27–40 × 12–20 µm) and confluentic acid as secondary metabolites [32]. In addition, *Myriostigma hainana* is similar to *Cryptothecia odishensis* R. Bajpai, S. Joseph & Upreti*,* but the latter has an I+ blue medulla, larger ascospores (105–142 × 35–52 µm) and barbatic acid and gyrophoric acid as secondary metabolites [33].

Phylogenetically, *Myriostigma hainana* is clustered with *Myriostigma miniatum* (Vain. ex Lücking) Aptroot, Ertz, Grube & M. Cáceres. They both have 8-spored asci, but *M. miniatum* has dark grey prothallus, bright orange ascigerous areas and 2-*O*-demethylperlatolic acid and orange pigment as secondary metabolites [34].

Additional specimens examined: CHINA, Hainan Province, Wenchang City, Dongjiao Town, Beigang Village, 19°32′46.30″ N, 110°51′10.44″ E, alt. 26 m, on the bark of trees, 7 March 2023, J.X. Xue & Y.M. Zhang 20230050 (SDNU); ibid., 20230051 (SDNU); ibid., Prima Resort, 19°31′51.02″ N, 110°50′54.14″ E, alt. 26 m, on the bark of trees, 6 March 2023, J.X. Xue & Y.M. Zhang 20230242 (SDNU).

**Figure 3 jof-10-00274-f003:**
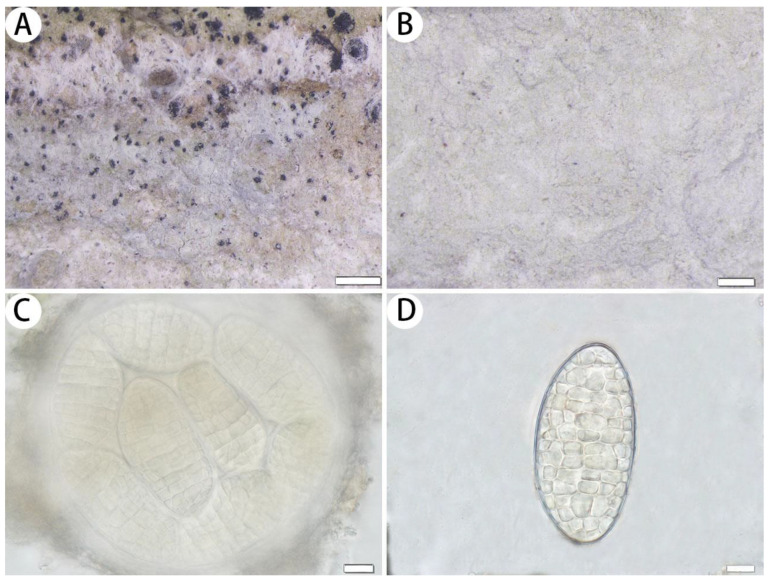
*Myriostigma hainana* (SDNU 20230061, holotype for (**A**,**C**,**D**); SDNU 20230051, type for (**B**)). (**A**) Thallus and prothallus. (**B**) Ascigerous areas. (**C**) Asci. (**D**) Ascospores. Scale bars: 500 µm (**A**), 200 µm (**B**), 10 µm (**C**,**D**).

*Myriostigma laxipunctata* J.X. Xue & Lu L. Zhang, sp. nov. (Figure 4)

MycoBank No: 850046

Diagnosis: The new species differs from other species of *Myriostigma* with its ascigerous areas having loose brown dots, slightly raised above the level of the thallus; hyaline ascospores (57–78 × 24–33 µm); gyrophoric acid and confluentic acid as secondary metabolites.

Type: CHINA, Yunnan Province, Jinghong City, Mengla County, Menglun Town, Xishuangbanna Tropical Botanical Garden, 21°55′43.17″ N, 101°15′39.08″ E, alt. 525 m, on the bark of trees, 5 March 2023, L.L. Liu et al. 20231052 (SDNU, holotype).

GenBank accession numbers: SSU PP051266, PP051265; RPB2 PP109369, PP109368; LSU PP033944, PP033943.

Description: Thallus corticolous, up to 4.5 cm in diameter, ecorticate, cottony, dull, pale green, firmly attached to the substrate, 75–90 µm thick. Isidia not observed. Prothallus usually distinct, thin, whitish byssoid, mainly composed of interwoven hyphae, 1–2 mm wide, forming a dark brown line while bordering different species. Medulla white, with calcium oxalate crystals. Photobiont trentepohlioid, cells rounded to oblong, single or a few cells aggregated, 7–10 × 5–8 µm. Hyphae 1–2.5 µm wide.

Ascigerous areas generally delimited, erumpent, developing in the thallus centre, slightly raised above the thallus level, pale greenish and with loose brown dots indicating individual asci, 140–175 µm thick. Asci hyaline, fragile, globose to subglobose, often covered by hyaline hyphae, 8-spored, 95–124 × 93–119 µm. Ascospores hyaline, oblong, muriform, straight or curved, often broader in the centre, 57–78 × 24–33 µm.

Pycnidia not observed.

Chemistry: thallus C+ red, K–, P–, UV+ pale grey-white; medulla and paraphysoids I+ sky-blue. TLC: gyrophoric acid and confluentic acid.

Etymology: The epithet refers to the ascigerous areas of the thallus that have loose brown dots.

Ecology and distribution: This species is found only in China on the bark of trees in a humid tropical forest in Yunnan Province.

Notes: Morphologically, *Myriostigma laxipunctata* is similar to *Myriostigma irregulare* (Lücking, Aptroot, Kalb & Elix) Frisch & G. Thor, but the latter has radiately elongated ascigerous areas, short-stalked asci and psoromic acid, subpsoromic acid, 2′-*O*-demethylpsoromic acid and confluentic acid as secondary metabolites [31]. In addition, *Myriostigma laxipunctata* is similar to *Cryptothecia stockeri* G. Neuwirth & Aptroot in having a firmly attached thallus and brown punctate ascigerous areas, but *C. stockeri* has a brown linear shaped prothallus, smaller ascospores (25–40 × 12–15 µm) and psoromic acid as a secondary metabolite [35].

Phylogenetically, *Myriostigma laxipunctata* is clustered with *Myriostigma flavescens* J.X. Xue & Lu L. Zhang and *M. candidum* Kremp. They both have 8-spored asci, but *M. flavescens* has white erumpent ascigerous areas with dense brown dots and yellow ascospores, *M. candidum* has smaller ascospores (40–65 × 12–25 µm) and 2′-*O*-methylperlatolic acid and 2′-*O*-methylanziaic acid as secondary metabolites [31].

Additional specimens examined: CHINA, Yunnan Province, Jinghong City, Mengla County, Menglun Town, Xishuangbanna Tropical Botanical Garden, 21°53′25.66″ N, 101°16′9.91″ E, alt. 592 m, on the bark of trees, 5 March 2023, L.L. Liu et al. 20231231 (SDNU); ibid., 21°55′30.14″ N, 101°15′42.72″ E, alt. 527 m, on the bark of trees, 5 March 2023, L.L. Liu et al. 20231124 (SDNU).

**Figure 4 jof-10-00274-f004:**
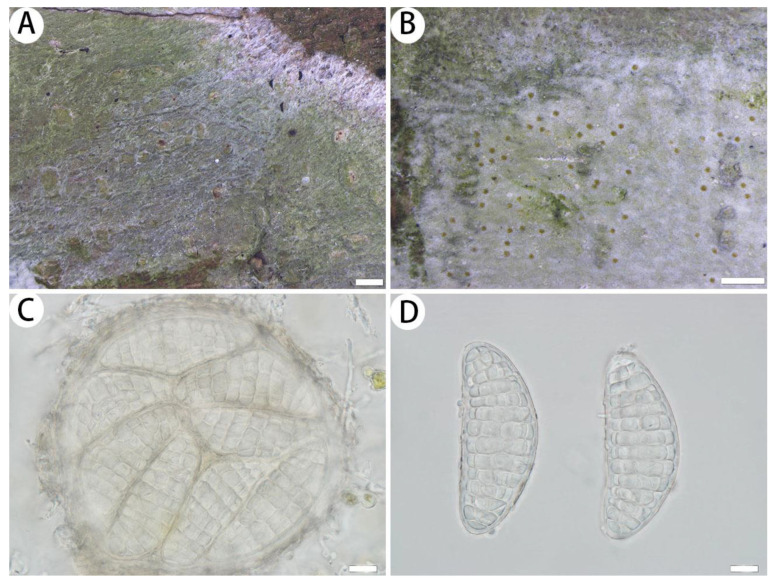
*Myriostigma laxipunctata* (SDNU 20231052, holotype). (**A**) Thallus and prothallus. (**B**) Ascigerous areas. (**C**) Asci. (**D**) Ascospores. Scale bars: 500 µm (**A**,**B**), 10 µm (**C**,**D**).

#### 3.2.2. The New Records

*Cryptothecia bartlettii* G. Thor, Symb. Bot. Ups. 32:1, 277 (1997) (Figure 5)

GenBank accession numbers: SSU PP051262, PP051261.

Description: Thallus corticolous, up to 7 cm in diameter, ecorticate, thick, cottony, dull, greenish grey to greyish white, loosely attached to the substrate, with paler, sometimes slightly raised areas, 175–260 µm thick. Isidia not observed, but the thallus had globose isidia-like structures (it does not contain cortex, but isidia contains cortex). Prothallus usually distinct, thick, whitish byssoid, mainly composed of interwoven hyphae, 0.5–1.5 mm wide. Medulla white, with calcium oxalate crystals. Photobiont trentepohlioid, cells rounded to oblong, single or aggregate into bundles, 9–15 × 7–12 µm. Hyphae 1–2 µm wide.

Ascigerous areas generally delimited, developing in the thallus centre, generally covered with globose isidia-like structures. Asci frequent, hyaline, ellipsoid to oblong, with a stalk, weakly aggregated, often covered by hyaline hyphae, 1-spored, 100–138 × 48–64 µm. Ascospores hyaline, ovoid to oblong, muriform, often slightly constricted just below the middle, surrounded by abundant cytoplasm in the asci when mature, (49–)68–100(–105) × (18–)23–36(–42) µm.

Pycnidia not observed.

Chemistry: thallus C+ red, K–, P–, UV+ pale grey-white; medulla and paraphysoids I+ sky-blue. TLC: gyrophoric acid.

Ecology and distribution: *Cryptothecia bartlettii* has previously been reported in New Zealand and Australia [2] and is new to China. We collected the species in Fujian Province under similar ecological conditions as in New Zealand and Australia. Specifically, we found them on the bark of trees within the subtropical forest at an elevation of approximately 450 to 570 m.

Notes: According to Thor [2], *Cryptothecia bartlettii* is characterized by a thick and loosely attached thallus, ascigerous areas covered by globose isidia-like structures and large ascospores [(49–)68–100(–105) × (18–)23–36(–42) µm]. The Chinese collections agree with this description, except for the longer asci (100–138 × 48–64 µm vs. 90–105 × 50–64 µm) and larger ascospores [(49–)68–100(–105) × (18–)23–36(–42) µm vs. (53–)69–84(–96) × (21–)25–30(–33) µm]. *Cryptothecia bartlettii* is similar to *C. eungellae* G. Thor in having a grey greenish thallus and globose isidia-like structures, but *C. eungellae* has a firmly attached thallus, shorter ascospores (55–72 µm long) and gyrophoric acid and norstictic acid as secondary metabolites [11].

Phylogenetically, *Cryptothecia bartlettii* is clustered with *C. inexspectata* G. Thor and *C. subnidulans* Stirt.; they all have 1-spored asci, but *C. inexspectata* has radiate and elongated whitish ascigerous areas and smaller ascospores (33–50 × 16–22 µm), and *C. subnidulans* has a firmly attached thallus, psoromic acid as a secondary metabolite and the ascigerous area without globose isidia-like structures [2].

Specimens examined: CHINA, Fujian Province, Fuzhou City, Jinan District, Gushan Town, Mt. Gushan, 26°03′22.99″ N, 119°23′16.10″ E, alt. 458 m, on the bark of trees, 13 July 2022, Lu L. Zhang & Q.J. Zuo 20220280 (SDNU); ibid., 26°03′43.78″ N, 119°23′14.49″ E, alt. 572 m, on the bark of trees, 13 July 2022, Lu L. Zhang & Q.J. Zuo 20220297 (SDNU); ibid., 26°03′19.51″ N, 119°23′10.64″ E, alt. 446 m, on the bark of trees, 13 July 2022, Lu L. Zhang & Q.J. Zuo 20220275 (SDNU).

**Figure 5 jof-10-00274-f005:**
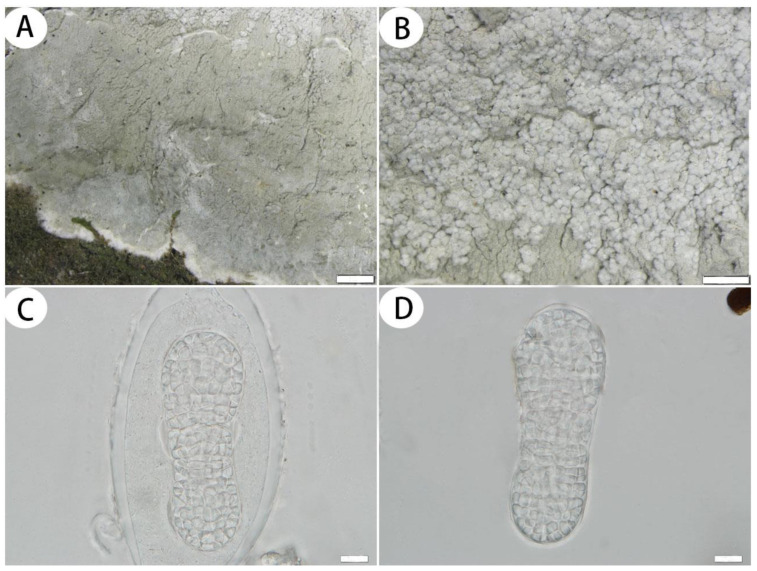
*Cryptothecia bartlettii* (SDNU 20220280). (**A**) Thallus and prothallus. (**B**) Ascigerous areas. (**C**) Asci. (**D**) Ascospores. Scale bars: 1 mm (**A**), 500 µm (**B**), 10 µm (**C**,**D**).

*Cryptothecia inexspectata* G. Thor, Symb. Bot. Ups. 32:1, 283 (1997) (Figure 6)

GenBank accession numbers: SSU PP051263, PP051264; RPB2 PP109371, PP109371.

Description: Thallus corticolous or bambusicolous, up to 4.5 cm in diameter, ecorticate, cottony, dull, greenish grey to whitish grey, sometimes yellowish green, firmly attached to the substrate, 85–100 µm thick. Isidia not observed. Prothallus usually distinct, thin, whitish byssoid, mainly composed of interwoven hyphae, 1–2 mm wide. Medulla white, with calcium oxalate crystals. Photobiont trentepohlioid, single or a few cells aggregated, cells rounded to oblong, 8–17 × 7–14 µm. Hyphae 1–3 µm wide.

Ascigerous areas generally delimited, whitish and slightly raised above the thallus level, irregular in outline and usually radiately elongated, aggregated in the thallus centre. Asci hyaline, oblong to obovate, with a stalk, weakly aggregated, often covered by hyaline hyphae, 1-spored, 41–60 × 31–40 µm. Ascospores hyaline, ovoid, muriform, often slightly constricted just below the middle, surrounded by abundant cytoplasm in the asci when mature, 33–50 × 16–22 µm.

Pycnidia not observed.

Chemistry: thallus C+ red, K–, P–, UV+ pale grey-white; medulla and paraphysoids I+ sky-blue. TLC: gyrophoric acid.

Ecology and distribution: *Cryptothecia inexspectata* has previously been reported in Australia, Papua New Guinea and Java [2,8] and is new to China. We collected the species in Fujian and Yunnan Provinces under similar ecological conditions as in Australia, Papua New Guinea and Java. Specifically, we found them on the bark of trees and on bamboo within the subtropical forest and tropical bamboo forest at an elevation of approximately 360 to 690 m.

Notes: According to Thor [2], Aptroot and Spier [8], *Cryptothecia inexspectata* is characterized by a firmly attached thallus and radiately elongated whitish ascigerous areas. The Chinese collections agree with this description, except for the shorter asci (41–60 µm long vs. 60–80 µm long). *Cryptothecia inexspectata* is similar to *C. striata* G. Thor in having radiately elongated whitish ascigerous areas, but *C. striata* has granular isidia and larger ascospores [(46–)55–70(–80) × (19–)23–29(–37) µm] [36].

Specimens examined: CHINA, Guangxi Province, Liuzhou City, Rongshui County, Qingshui Pond Protection Station, 25°11′59.79″ N, 108°47′46.47″ E, alt. 359 m, on the bark of trees, 19 December 2020, L. Hu et al. 20200434 (SDNU); ibid., 20200378 (SDNU); Yunnan Province, Xishuangbanna Dai Autonomous Prefecture, Jinghong City, Primitive Forest Park, 22°1′55.75″ N, 100°52′37.47″ E, alt. 689 m, on bamboo, 7 March 2023, L.L. Liu et al. 20230639 (SDNU); ibid., 22°2′9.71″ N, 100°53′5.81″ E, alt. 716 m, on the bark of trees, 7 March 2023, L.L. Liu et al. 20230668 (SDNU).

**Figure 6 jof-10-00274-f006:**
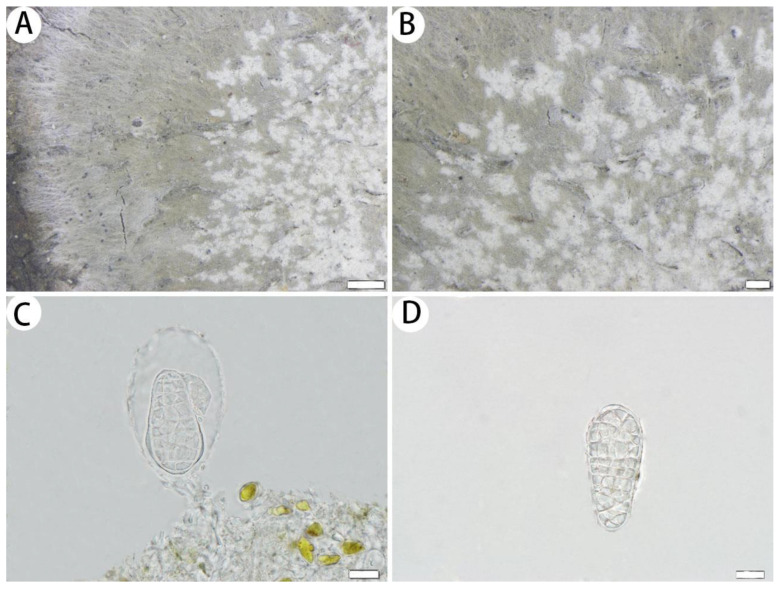
*Cryptothecia inexspectata* (SDNU 20200434 for (**A**,**B**); SDNU 20230639 for (**C**,**D**)). (**A**) Thallus and prothallus. (**B**) Ascigerous areas. (**C**) Asci. (**D**) Ascospores. Scale bars: 500 µm (**A**), 200 µm (**B**), 10 µm (**C**,**D**).

Key to the *Cryptothecia* and *Myriostigma* species occurring in China
1a.Asci 1–2-spored—21b.Asci 8-spored—42a.Thallus P+ yellow; with psoromic acid—*C. subnidulans*2b.Thallus P–; without psoromic acid—33a.Ascigerous areas whitish and usually radiately elongated; ascospores 33–50 × 16–22 µm—***C. inexspectata***3b.Ascigerous areas generally covered with globose isidia-like structures; ascospores (49–)68–100(–105) × (18–)23–36(–42) µm—***C. bartlettii***4a.Thallus without lichen substances—54b.Thallus with lichen substances—65a.Ascospores narrow; 60–76 × 17–30 µm—*C. aleurella*5b.Ascospores broad; 65–108 × 42–50 µm—*C. aleurocarpa*6a.Thallus C+ red; with gyrophoric or 2′-*O*-methylanziaic acids—76b.Thallus C–; without gyrophoric or 2′-*O*-methylanziaic acids—107a.Thallus with gyrophoric acid and additional substances—87b.Thallus with 2′-*O*-methylanziaic and 2′-*O*-methylperlatolic acids; ascospores 40–65 × 12–25 µm—*M. candidum*8a.Ascigerous areas indistinct; additionally with methyl 2′-*O*-methylmicrophyllinate; ascospores 52–88 × 24–47 µm—***M. hainana***8b.Ascigerous areas distinct; additionally with confluentic acid—99a.Ascigerous areas white with dense brown dots; ascospores yellow; 58–76 × 19–28 µm—***M. flavescens***9b.Ascigerous areas pale greenish with loose brown dots; ascospores hyaline; 57–78 × 24–33 µm—***M. laxipunctata***10a.Thallus P+ yellow; with psoromic acid; ascospores 50–70 × 30–37 µm—*C. polymorpha*10b.Thallus P–; without psoromic acid; ascospores 27–40 × 12–20 µm—*C. subtecta*

## 4. Discussion

As one of the largest families of lichens, Arthoniaceae contains more than 700 lichenized species according to Lücking et al. [37]. Recent phylogenetic analyses by Frisch et al. [7], Thiyagaraja et al. [38] and others have further implications for our understanding of generic limits within this family. However, it should be noted that all of these analyses were based on only a relatively low number of species described in Arthoniaceae. In previous studies, we can see that the morphological characteristics of *Cryptothecia* and *Myriostigma* are very similar, and some species of *Myriostigma* were described as *Cryptothecia.* In 2014, Frisch et al. [7] revealed through phylogenetic analysis that *Cryptothecia* was polyphyletic and *Myriostigma* Krempelhuber (1874) was resurrected because the type species *Myriostigma candidum* Kremp. [=*Cryptothcia candida* (Kremp.) R. Sant.] represents a lineage separate from *Cryptothecia*. They also reinstated *Myriostigma* for the *Cryptothecia candida* group, which is characterized by raised maculate ascomata-like ascigerous areas; 8-spored, thick-walled, globular and stalked asci; and bean-shaped, muriform ascospores [7]. Therefore, some species of *Cryptothecia,* such as *C. miniata* and *C. napoensis,* were transferred to *Myriostigma* [10,34]. However, in our study, we can see that some species of *Myriostigma,* such as *M. hainana* and *M. laxipunctata,* resemble *Cryptothecia* in morphology due to the absence of raised maculate ascomata-like ascigerous areas; thus, the distinction between the two genera based on the presence or absence of a maculate ascomata-like ascigerous areas becomes ambiguous. In addition, in our research, all species of *Cryptothecia* formed a cluster with the type species, *C. subnidulans* Stirt., on the phylogenetic trees*,* which exhibited a consistent presence of 1–2-spored asci, and all species of *Myriostigma* have 8-spored asci. The number of ascospores may thus serve as a distinguishing characteristic between *Cryptothecia* s. str. and *Myriostigma*. The confirmation of this viewpoint requires further research.

Species in Arthoniaceae often have a pantropical distribution. In southern China, there are abundant subtropical to tropical evergreen resources [39]. This habitat is favourable for the lichens of Arthoniaceae. However, the family has not been sufficiently studied; thus far, only a few species have been recorded in China, including five species of *Cryptothecia* and one species of *Myriostigma* [40]. In our study, we found that Arthoniaceae is highly diverse in southern China, especially in the Xishuangbanna region of Yunnan Province. Therefore, more comprehensive sampling and research are needed to determine the true diversity of Arthoniaceae in southern China.

## Figures and Tables

**Figure 1 jof-10-00274-f001:**
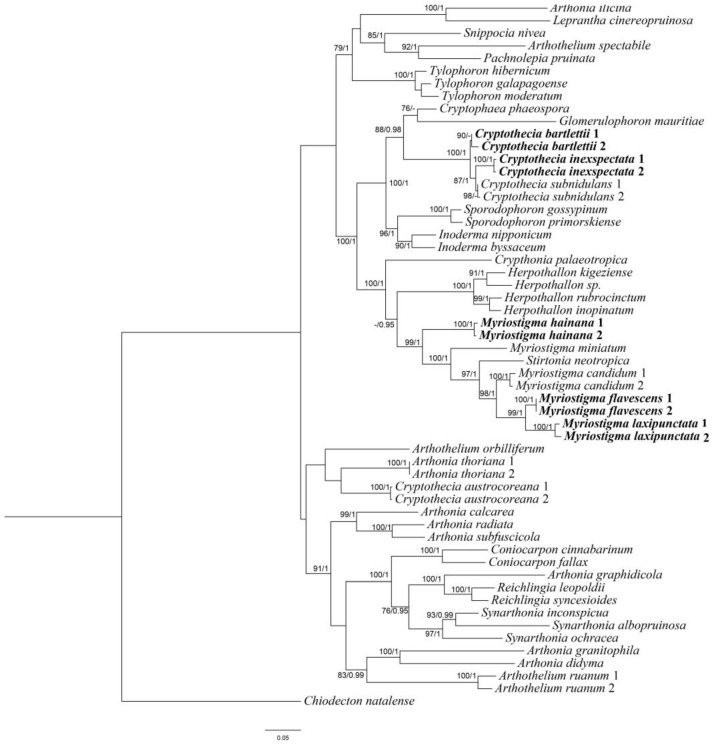
Phylogenetic tree constructed by maximum likelihood (ML) analysis of Arthoniaceae species based on the concatenated mtSSU-RPB2-nLSU dataset. Bootstrap support values ≥ 70 for ML and posterior probabilities ≥ 0.95 (second value) for Bayesian methods are indicated above or below the branches. Newly obtained sequences are marked in bold.

**Table 1 jof-10-00274-t001:** Specimens used for the phylogenetic analyses with the corresponding voucher information and GenBank accession numbers for the mtSSU, RPB2 and nLSU sequences. Newly obtained sequences in this study are in bold.

Species Name	Voucher Specimen	GenBank Accession Number		
		mtSSU	RPB2	nLSU
*Arthonia calcarea*	Thor 11/6a (UPS)	KJ850974	KJ851105	–
*Arthonia didyma*	Ertz 7587 (BR)	EU704047	EU704010	EU704083
*Arthonia granitophila*	Frish 10/Se74 (UPS)	KJ850981	KJ851107	KJ851049
*Arthonia graphidicola*	Frisch 10/Jp102 (UPS)	KJ850980	–	KJ851034
*Arthonia ilicina*	McCune 31067	KJ850982	–	–
*Arthonia radiata*	Frisch 10/Se29 (UPS)	KJ850968	KJ851108	–
*Arthonia subfuscicola*	Thor 11/1 (UPS)	KJ850971	KJ851110	–
*Arthonia thoriana 1*	Sanderson 2176 (BR)	MG207687	–	–
*Arthonia thoriana 2*	Sanderson 2174 (herb.Sanderson)	MG207685	–	–
*Arthothelium orbilliferum*	TRH-L-15449	KY983977	–	–
*Arthothelium ruanum 1*	KoLRI 038018	MF616609	MF616619	–
*Arthothelium ruanum 2*	KoLRI 038261	MF616611	MF616621	–
*Arthothelium spectabile*	Frisch 12Jp179a (TNS)	KP870144	KP870160	–
*Chiodecton natalense*	Ertz 6576 (BR)	EU704051	EU704014	EU704085
*Coniocarpon cinnabarinum*	Johnsen 111003 (UPS)	KJ850976	KJ851103	KJ851083
*Coniocarpon fallax*	LD:L10075	KJ850979	KJ851101	–
*Crypthonia palaeotropica*	Frisch 11/Ug457 (UPS)	KJ850961	KJ851084	–
*Cryptophaea phaeospora*	Van den Broeck 5809 (BR)	KX077541	–	–
*Cryptothecia austrocoreana 1*	KolRI No. 041892	MF769375	–	–
*Cryptothecia austrocoreana 2*	KolRI No. 044721	MF769374	–	–
** *Cryptothecia bartlettii 1* **	**Zhang et al. 20220297 (SDNU)**	**PP051262**	–	–
** *Cryptothecia bartlettii 2* **	**Zhang et al. 20220275 (SDNU)**	**PP051261**	–	–
** *Cryptothecia inexspectata 1* **	**Liu et al. 20230668 (SDNU)**	**PP051263**	**PP109371**	–
** *Cryptothecia inexspectata 2* **	**Liu et al. 20230639 (SDNU)**	**PP051264**	**PP109370**	–
*Cryptothecia subnidulans 1*	v.d.Boom 40613 (hd v.d. Boom)	KJ850952	KJ851087	–
*Cryptothecia subnidulans 2*	Joensson Guyana 6a (UPS)	KJ850953	KJ851088	–
*Glomerulophoron mauritiae*	Ertz 19164 (BR)	KP870153	KP870167	–
*Herpothallon inopinatum*	Rudolphi 12 (UPS)	KJ850964	KJ851099	–
*Herpothallon kigeziense*	Frisch 11/Ug26 (UPS)	KF707644	KF707654	–
*Herpothallon rubrocinctum*	Rudolphi 5 (UPS)	KF707643	KF707655	–
*Herpothallon sp.*	Frisch 11/Ug401 (UPS)	KF707645	KF707653	–
*Inoderma byssaceum*	Thor 25952 (UPS)	KJ850962	KJ851089	KJ851040
*Inoderma nipponicum*	Frisch 12Jp227 (TNS)	KP870146	KP870162	–
*Leprantha cinereopruinosa*	Kukwa 17127 & Lubek (BR)	MG207692	–	–
*Myriostigma candidum 1*	Ertz 9260 (BR)	EU704052	EU704015	HQ454520
*Myriostigma candidum 2*	Frisch 11/Ug125 (UPS)	KJ850959	KJ851096	–
** *Myriostigma flavescens 1* **	**Liu et al. 20230612 (SDNU)**	**PP051268**	**PP130144**	–
** *Myriostigma flavescens 2* **	**Liu et al. 20230641 (SDNU)**	**PP051267**	–	–
** *Myriostigma hainana 1* **	**Xue et al. 20230061 (SDNU)**	**PP051271**	**PP101845**	–
** *Myriostigma hainana 2* **	**Xue et al. 20230050 (SDNU)**	**PP051272**	**PP109365**	–
** *Myriostigma laxipunctata 1* **	**Liu et al. 20231231 (SDNU)**	**PP051266**	**PP109369**	**PP033944**
** *Myriostigma laxipunctata 2* **	**Liu et al. 20231052 (SDNU)**	**PP051265**	**PP109368**	**PP033943**
*Myriostigma miniatum*	Silva T2A29 (ISE—epitype)	KP843606	–	–
*Pachnolepia pruinata*	Frisch 11/Se34 (UPS)	KJ850967	KJ851098	–
*Reichlingia leopoldii*	Ertz 13293	JF830773	HQ454722	HQ454581
*Reichlingia syncesioides*	Frisch 11/Ug14 (UPS)	KF707651	KF707656	KF707636
*Snippocia nivea*	Ertz 17437 (BR)	MG207695	–	–
*Stirtonia neotropica*	Cáceres & Aptroot 11112 (ISE)	KP843611	–	–
*Sporodophoron gossypinum*	Frisch 12Jp186 (TNS)	KP870154	KP870168	–
*Sporodophoron primorskiense*	Ohmura 10509 (TNS)	KP870157	KP870169	–
*Synarthonia albopruinosa*	VDB 6086 (BR<BEL>)	MH251873	–	–
*Synarthonia inconspicua*	VDB 7013B (BR<BEL>)	MH251881	–	–
*Synarthonia ochracea*	VDB 6653 (BR<BEL>)	MH251884	–	–
*Tylophoron galapagoense*	Bungartz 8749 (CDS)	JF830776	–	JF295078
*Tylophoron hibernicum*	Frisch 11/Ug220 (UPS)	KJ850966	KJ851097	KJ851065
*Tylophoron moderatum*	Ertz 14504 (BR)	JF830780	–	JF295085

## Data Availability

The names of the new species were formally registered in the MycoBank database (https://www.mycobank.org/, accessed on 16 September 2023). Specimens were deposited in the Lichen Section of the Botanical Herbarium, Shandong Normal University (SDNU). The newly generated sequences were deposited in GenBank (https://www.ncbi.nlm.nih.gov/genbank, accessed on 24 December 2023).

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
