# Peer review of "The Phylogeny and Taxonomy of Cryptothecia (Arthoniaceae, Ascomycota) and Myriostigma (Arthoniaceae, Ascomycota), including Three New Species and Two New Records from China"

_jof, 2024, doi:10.3390/jof10040274_

Round 1

Reviewer 1 Report (Previous Reviewer 2)

Only minor comments (see below)

Line 140-141: ” two new Chinese records species” should be ” two new Chinese records” as both are clustered with Cryptothecia subnidulans.

Line 142: I do not understand what the words “which supported the delimitation of them”.

Line 187: “froming” should be ”forming”

Line 214: “They all have” should be “They both have”

Line 218: “Therefore, we introduce M. hainana as a new species.” should be deleted.

Line 242: “froming” should be ”forming”

Line 272: “They all” should be ”They both”.

Line 277; “Therefore, we introduce M. laxipunctata as a new species.” should be deleted.

Line 345: “linear shape” should be ”linear shaped”

Line 350: “1–5mm” should be ”1–5 mm”

Line 398: “32:” should be ”32:1,”

Line 403: ”(It dose not contain cortex, but isidia contains cortex.)”. The sentence is not correct English, “dose” should be “does” and do not use “.” in a parentheses.

Line 426-427: “which may be due to the different ecological and geographical conditions” should be deleted, just speculations.

Line 447: ” 32:” should be ”32:1,”

Other proposals

1. Please present the new species in alphabetical order.

2. The key is still not good. Under “Discussion” the authors write “Besides, in our research, all species of Cryptothecia formed a cluster with the type species, C. subnidulans Stirt., on the phylogenetic trees, of which exhibited a consistent presence of 1–2-spored asci; and all species of Myriostigma have 8-spored asci. The number of ascospores may thus serve as a distinguishing characteristic between Cryptothecia s. str. and Myriostigma need”.  It would therefore be much logical to start the key using the number of spores in the asci. I strongly recommend the authors to rewrite the key.

3. Among the references, some journals are shortened as “Int. J. Pl. Sci.” while other are written in full like “Molecular Ecology Resources”. Write journals in a consistent way.  

Author Response

Thank you very much for taking the time to review this manuscript. Based on your opinion, I have made revisions to my article.

Response to detail comments:

  1. Line 147-148: “two new Chinese records species”has been changed to “two new Chinese records as both are clustered with Cryptothecia subnidulans.”
  2. Line 149: “which supported the delimitation of them”has been changed to “which supported the delimitation of two new Chinese records.”
  3. Line 216: “froming” has been changed to “forming”.
  4. Line 240: “They all have”has been changed to “They both have”
  5. Line 242: “Therefore, we introduce hainana as a new species.” has been deleted.
  6. Line 266: “froming” has been changed to “forming”.
  7. Line 290: “They all have” has been changed to “They both have”
  8. Line 293: “Therefore, we introduce  laxipunctata as a new species.” has been deleted.
  9. Line 173: “linear shape” has been changed to “linear shaped”
  10. Line 177: “1–5mm” has been changed to “1–5 mm”
  11. Line 303: “32:” has been changed to“32:1,”
  12. Line 308: “(It dose not contain cortex, but isidia contains cortex.).” has been changed to“ (It does not contain cortex, but isidia contains cortex). ”
  13. Line 332: “which may be due to the different ecological and geographical conditions” has been deleted.
  14. Line 351: “32:” has been changed to“32:1,”

Response to other proposals:

  1. The new species have been modified in alphabetical order.
  2. The key has been modified.
  3. The reference section has changed the journal to its full name.

Reviewer 2 Report (New Reviewer)

The manuscript describes Chinese Cryptothecia and Myriostigma species presenting three new species for sciences and new distribution records of two species for the country.

The Discussion contains important comments on the necessity for further research.

The phylogeny and taxonomy of Cryptothecia (Arthoniaceae, Ascomycota) and Myriostigma (Arthoniaceae, Ascomycota), including three new species and two new records from China

The manuscript describes Chinese Cryptothecia and Myriostigma species presenting three new species for sciences and new distribution records of two species for the country.

Illustration tables, cladogram and a key to species are provided.

Most of the microscopic figures need more contrast and less light on the translucent ascospores mostly for the better visibility of the septation.

It is not explained why the otherwise widely used ITS was not analysed. It is usually regarded to be important in the separation of species and suggested as a standard in fungal taxonomy, but there are also taxonomic groups (e.g. in Cladoniaceae) where the application of ITS alone is insufficient.

It is clear that authors followed Frisch et al. 2014 where the same markers were applied.

Still, there should be some text about clarifying why the sequences were chosen.

Diagnosis should present differences from the already existing species: „The new species differs from other species of Myriostigma by ………….”

The Discussion contains important comments on the necessity for further research.

The English text is well written with hardly any mistakes.

Some minor items are indicated directly in the manuscript pdf file.

Author Response

Thank you very much for taking the time to review this manuscript. Based on your opinion, I have made revisions to my article.

  1. TLC operation methods has been improved, and the description of other methods is similar to other literature in the journal.
  2. The English languagehas been appropriately edited.
  3. The micrograph has been modified to make the ascosporesseptation clearer.
  4. Line 90-93: I have provided the reason for not using ITS.
  5. Line 164 (and other places): “This new species is characterized by”has been changed to “The new species differs from other species of Myriostigma by ………….”
  6. The error in pdf file has been corrected.

Reviewer 3 Report (New Reviewer)

This is a well-written work concerning taxonomy of the Arthoniaceae family, which includes description of three new species of Myriostigma genus discovered. The research and analyses were carried out correctly, and the taxon descriptions were prepared accurately. That's why I don't have many comments.

1. In my opinion, the abstract does not need to contain diagnostic features of the discovered species.

2. Line 330. Are the ecological and habitat conditions in the mentioned sites (New Zealand and Australia) similar to those in China? Can you give more details?

3. Line 375. Are the ecological and habitat conditions in the mentioned sites (Australia, Papua New Guinea and Java) similar to those in China?

Author Response

Thank you very much for taking the time to review this manuscript.  

Response to detail comments:

  1. I have revised the abstract section based on your suggestions.
  2. Line 323 to Line 326: The growth environment of  C. bartlettii in China is similar to New Zealand and Australia, and its growth environment is described.
  3. Line 369 to Line 373: The growth environment of  C. inexspectata in China is similar to Australia, Papua New Guinea and Java, and its growth environment is described.

This manuscript is a resubmission of an earlier submission. The following is a list of the peer review reports and author responses from that submission.

Round 1

Reviewer 1 Report

Comments and Suggestions for Authors

Well, I have to reject this manuscript because the phylogeny and presented taxonomy does not correspond at all. It is not allowed to use paraphyletic taxa. You either have to do extensive nomenclatural rearrengements by enlargening concepts of Cryprothecia (by including Myriostigma) or include some of your species to Arthonia s. str. by enlarging concept of Arthonia. 

The quality of the microphotos should be improved because it is impossible to see the diagnostic characteristics. 

Why did you not use fungal barcoding marker? The material and methods should be more detailed, as well as species descriptions. The species Diagnosis is the most important part of the description that says what are the most important characteristics that separate it from the most similar / close taxon. 

Author Response

Thank you very much for taking the time to review this manuscript. Now, I based on your suggestions and the phylogenetic tree in this article, classified the new species, such as defining species Cryptothecia hainana as Myriostigma hainana and Cryptothecia laxipunctata as Myriostigma laxipunctata, and deleted Cryptothecia albus-striata and Cryptothecia faveomaculata. At the same time, corresponding modifications have been made to the table 1., figure 1., methods, results and discussion sections of the article.

The quality of the microphotos has been improved.

The method used in this article is based on reliable and practical experience obtained from previous literature, therefore it was not adopted fungal barcoding marker.

The material and methods has been improved, and TLC operation methods have been added.

The species descriptions also has been improved, but due to the relatively simple structure of Cryptothecia and Myriostigma, the space for improvement is limited, I only adding the thickness of the thallus, the width of the prothallus, and the shape of the photobiont.

Reviewer 2 Report

Comments and Suggestions for Authors

The manuscript is well written and include new and interesting results. I support the authors conclusion to wait with splitting Cryptothecia and Myriostigma in monophyletic groups until more molecular data is available. The authors should consider to cooperate with researchers in other countries in order to get access to material of more species to sequence.

Major comments

1. The English need improvement and must be checked.

2. The authors use both the terms “hyaline” and colourless”. Please define the difference or use only one of the terms.

3. The authors name a new species Cryptothecia albus-striata. However, according to Latin grammatics, the new species name should be written “albostriata”.

3. In case the typematerial is enough, it would be great to place isotype material in another herbarium.

4. The key is confusing and not possible to use.  To improve it (A) only include characters that differ, not “thick-walled” for both alternatives, (2) key out all species having one character before starting with the next alternative, e.g. key out all species with 1-spored asci before start to key out species with 8-spored asci, (3) consider to write “>” instead of “over” and “<” instead of “up to”.  

Minor comments

Line 137-139: Indicate if there is any character which is only found in Cryptothecia austrocoreana and C. albostriata, supporting that they are not closely related to other Cryptothecia (and should be placed in the Arthonioid clade).

Line 166-167: The wording “(insoluble in KOH, forming colourless, needle-shaped crystals in 25% H2SO4)” is not necessary to include under the species descriptions. Instead write it under “Material and Methods (part of this text is already here).

Line 167-168: Not necessary to repeat “yellowish green” for each species as there is no difference. Instead, write this under “Material and Methods”.

Line 175 (and other places): “Spot test:” can be deleted.

Line 183 (and other places): “C. subtecta has an I+ medulla”; indicate colour, e.g. “I+ red”.

Line 195 (and other places): instead of “same collection data for preceeding”, write “ibid.”

Line 226-227: Improve the description of the species epithet. “Laxi” means roughly “relaxed” or “loosely” but the Etymology is general and give no explanation why this epithet is used.

Line 272: “but” can be excluded.

Line 276-277 (and other places): “medulla and paraphysoids I+ blue”. Indicate if the blue colour is pale, sky-blue or dark blue.

Line 311 (and other places): “on the bark of bamboos”. Note that bamboo belong to the family Poaceae. which belong to Monocotelydones and this group does not have bark. Instead, just write “on bamboo”.

Line 338: write full genus names instead of “S.” and “C.”.

Line 386-387: Explain why the structures are “isidia-like” and not “isidia”.

Line 388: “many calcium oxalate crystals”. This is the only place where the abundance of the calcium oxalate crystals is indicated. Indicate the abundance for all species or for none of the species.

Line 438: “8–17×7–14”. Add empty spaces before and after “×”

Line 462: “Yunnan Province:” should be “Yunnan Province,”

Line 550: Avoid to start a sentence with “And”.

Line 567: It is unclear to me why “Lu” is not shortened to “L.” as first names are shortened for the other persons.

Line 587: “32” should be “32:1”

Line 614: Delete “British,”

Comments on the Quality of English Language

The English need improvement and must be checked.

Author Response

Response : Thank you very much for taking the time to review this manuscript. I have sent a request to relevant foreign researchers, but have not received any additional specimens or molecular sequences. In addition, based on the suggestions of foreign researchers and the phylogenetic tree proposed in this article, new species were classified, such as defining species Cryptothecia hainana as Myriostigma hainana and Cryptothecia laxipunctata as Myriostigma laxipunctata, and deleting Cryptothecia albostriata and Cryptothecia faveomaculata. At the same time, corresponding modifications have been made to the table 1., figure 1., methods, results and discussion sections of the article.

Response to main comments:

  1. The English has been modified.
  2. “colourless”has been deleted.
  3. This new species has been deleted.
  4. The number of specimens is small, so the specimensare only stored in the Lichen Section of Botanical Herbarium (SDNU, Shandong Normal University). 
  5. The key has been modified.

Response to minor comments:

  1. Cryptothecia albostriata has been deleted.
  2. The wording “(insoluble in KOH, forming colourless, needle-shaped crystals in 25% H2SO4)” has been deleted.
  3. I removed “yellowish green” from each species description and included it in the “introduction”, and I think it would be more appropriate to place it here.(Line 27)
  4. Line 173(and other places): “Spot test:” has been deleted.
  5. Line 181(and other places): “C. subtecta has an I+ medulla” has been changed to “ subtecta has an I+ blue medulla”.
  6. Line 193(and other places): “same collection data for preceeding” has been changed to “ibid.”.
  7. Line 224-225: I revisedthe description of the species epithet, and “sparse” has been changed to “loosely”.
  8. Line 169: “but” has been deleted. And Cryptothecia albostriata has been deleted.
  9. Line 222-223(and other places): “medulla and paraphysoids I+ blue”has been changed to ”medulla and paraphysoids I+ sky-blue”
  10. Line 257 (and other places): “on the bark of bamboos” has been changed to “on bamboo”.
  11. The “Line 338” part in the original text has been deleted.
  12. Line 315: Explained the reason why “isidia-like” is not “isidia”.
  13. Line 317: “many ”has been deleted.
  14. Line 364: “8–17×7–14”has been changed to “8–17 × 7–14”.
  15. Line 388: “Yunnan Province:” has been changed to “Yunnan Province,”.
  16. Line 431: “And”has been changed to “In 2014,”.
  17. Line 459: “L. Zhang” has already been used in China, and for ease of differentiation, “Lu-Lu Zhang” is abbreviated as “Lu L. Zhang”.
  18. Line 480-481: “32” has been changed to “32:1”.

    19. Line 512: “British,” has been deleted.